# Structural diversity of B-cell receptor repertoires along the B-cell differentiation axis in humans and mice

Aleksandr Kovaltsuk[1], Matthew I. J. Raybould[1], Wing Ki Wong[1], Claire Marks[1], Sebastian Kelm[2], James Snowden[2], Johannes Trück[3], Charlotte M. Deane[1]*

1 Department of Statistics, University of Oxford, Oxford, United Kingdom, 2 UCB Pharma, Slough, United Kingdom, 3 Division of Immunology, University Children's Hospital, University of Zurich, Zurich, Switzerland

* deane@stats.ox.ac.uk

**Data Availability Statement:** All data are within the manuscript and supporting information files, except for our software tool, which is available at https://github.com/oxpig/saab_plus under BSD 3-Clause license.

## Abstract

Most current analysis tools for antibody next-generation sequencing data work with primary sequence descriptors, leaving accompanying structural information unharnessed. We have used novel rapid methods to structurally characterize the complementary-determining regions (CDRs) of more than 180 million human and mouse B-cell receptor (BCR) repertoire sequences. These structurally annotated CDRs provide unprecedented insights into both the structural predetermination and dynamics of the adaptive immune response. We show that B-cell types can be distinguished based solely on these structural properties. Antigen-unexperienced BCR repertoires use the highest number and diversity of CDR structures and these patterns of naïve repertoire paratope usage are highly conserved across subjects. In contrast, more differentiated B-cells are more personalized in terms of CDR structure usage. Our results establish the CDR structure differences in BCR repertoires and have applications for many fields including immunodiagnostics, phage display library generation, and "humanness" assessment of BCR repertoires from transgenic animals. The software tool for structural annotation of BCR repertoires, SAAB+, is available at https://github.com/oxpig/saab_plus.

## Author summary

B-cell receptors (BCR) are the major components of the adaptive immune system. These are immunoglobulin molecules that bind to foreign substances known as antigens. Each individual has a huge BCR repertoire, where each individual BCR has a specific binding site composed of the complementary-determining regions (CDRs) capable of recognising a specific antigen. Drug discovery and immunodiagnostics inspired by the adaptive immune system rely on our ability to accurately interrogate the structural diversity of the binding sites of the BCR repertoire. Here we report our novel rapid pipeline, SAAB+, which has enabled us to interrogate how the structure of the CDR changes in BCR repertoires along the B-cell differentiation axis. By analysing human and mouse BCR repertoires at an unprecedented scale, we observed species-specific structural predetermination

**Funding:** AK is supported by funding from Biotechnology and Biological Sciences Research Council (BBSRC) [BB/M011224/1], UCB Pharma Ltd and Royal Commission for the Exhibition of 1851 Industrial Fellowship awarded to AK. MIJR and WKW are supported by Engineering and Physical Sciences Research Council and Medical Research Council Grant EP/L016044/1, GlaxoSmithKline plc, AstraZeneca plc, UCB Celltech, and F. Hoffmann-La Roche AG. The funders had no role in study design, data collection and analysis, decision to publish, or preparation of the manuscript.

**Competing interests:** The authors declare that the research was conducted in the absence of any commercial or financial relationships, with UCB Pharma or otherwise, that could be construed as a potential conflict of interest.

and detected CDR dynamics across multiple stages of B-cell differentiation. We showed that naïve repertoires share the highest number and diversity of CDR structures, a pattern which was highly conserved in all B-cell donors. Our results suggest that increased B-cell differentiation is associated with a personalization of CDR structure usages. Finally, we established the differences in CDR usages between humans and mice, analysis with immediate relevance for BCR repertoire "humanness" assessment and rational immunotherapeutic engineering.

## Introduction

B-cells are essential components of the adaptive immune system in jawed vertebrates. They play a key role in recognizing foreign molecules (antigens) via membrane-bound B-cell receptors (BCR), and antibodies (secreted BCRs). Successful recognition of a broad array of structural motifs (epitopes) on antigens relies on the enormous sequence and structural diversity of BCR repertoires, generated by the rearrangement of V(D)J gene segments in the two variable domain chains (heavy and light), each consisting of four framework and three complementary-determining (CDR) loop regions [1,2]. Upon antigen stimulation, somatic hypermutation (SHM) recursively introduces changes to the variable (Fv) domain of naïve BCR repertoires. These occur primarily in the antibody binding interface (paratope, which consists mostly of CDR residues)[3], leading to structural changes. Those B-cells whose paratopes are epitope-complementary are clonally expanded, and further diversified and selected to enhance antigen binding properties. BCR diversification also happens outside the Fv domain, where immunoglobulin class switching changes the constant region of the heavy chain [4]. There are five main heavy constant regions (isotypes), each with a unique profile of effector functions and antigen binding avidity.

Next-generation sequencing of immunoglobulin genes (Ig-seq) has become an essential technique in immunology [5,6]. For example, Ig-seq has revealed the dynamics of BCR sequence diversification across different B-cell types in healthy and antigen-stimulated B-cell donors [7–10], advanced our understanding of the adaptive immune response, and contributed to vaccine development [11] and immunodiagnostics [12].

Most Ig-seq analysis tools work within the remit of BCR primary sequence information [6,13]. These rapid methods of measuring BCR diversity are highly scalable, an important property as Ig-seq datasets become ever larger and more numerous [2]. However, the decision to avoid paratope structural descriptors could lead to inaccuracies [14–16], as it is known that similar sequences can have markedly different epitope complementarity and *vice versa* [13]. Therefore, a computationally-efficient structure-based BCR repertoire method should augment current Ig-seq analysis pipelines to deliver a clearer understanding of the process of BCR development.

One of the first structural analyses of Ig-seq data was that of DeKosky et al., [14]. They demonstrated that antibody models from paired-chain naïve and memory BCR repertoires displayed different physicochemical properties. However, their analysis was limited to 2,000 antibody models from three B-cell donors [14]. Most publicly-available BCR repertoires are unpaired, only covering either the heavy or light variable domain [17] precluding the generation of refined antibody models. Krawczyk et al., [15] showed that it was possible to annotate unpaired BCR repertoires with structural information by mapping loop sequences individually onto crystallographically-solved antibody structures.

Using a similar approach, we have investigated structural diversity along the B-cell differentiation axis in humans and mice. We show that structurally annotating BCR repertoires yields unprecedented insights into both the structural predetermination and dynamics of the adaptive immune response. By approximating BCR repertoire structures with rapid homology modelling techniques, we find that different B-cell types can be distinguished by their usage of CDR loop structures. Our analysis reveals that BCR repertoires of naïve B-cells tend to contain conserved "public" CDR structure profile, whilst those of more differentiated B-cell types become more personalized. These results provide crucial information about the structural changes in antibody CDRs during B-cell differentiation, with a plethora of prospective applications in immunodiagnostics and rational immunotherapeutic engineering.

## Results

### Structural annotation of Ig-seq data

We searched the Observed Antibody Space (OAS) resource [17] for heavy chain Ig-seq studies that contained at least three different B-cell types, had sequences with defined isotype information and consisted of at least 50 BCR repertoires, and identified two studies: Galson et al., ("human") [7] and Greiff et al., ("mouse") [9].

Annotating the antibody CDR sequences in these human and mouse Ig-seq studies with structural information allows us to investigate how the three-dimensional shape of CDR-H1, CDR-H2 and CDR-H3 loops vary across BCR repertoires (Fig 1). To achieve this, we developed the SAAB+ pipeline.

SAAB+ is our analysis pipeline that performs annotation of BCR repertoire amino acid data with structural information. The SAAB+ pipeline was built on the previously developed tool, SAAB [15]. The major differences/improvements of SAAB+ over SAAB are the following. First, SAAB+ scrutinises each antibody sequence for structural viability. In this step, sequences are passed through multiple structural filters (alignment to Hidden Markov Model (HMM) profiles, indel and conserved residue identification, chimeric sequence removal, presence of all 3 CDR loops according to the IMGT numbering [19] definition) [17,20]. SAAB+ utilizes SCALOP [21] for rapid canonical class identification, which gives a significant boost to the analysis rate without compromising accuracy. In a similar fashion to SAAB, FREAD is employed to predict whether CDR-H3s from the Ig-seq data share a similar structure to a crystallographically-solved CDR-H3 structure [22]. CDR-H3 predictions made by FREAD are annotated with the PDB code of the crystallographically-solved CDR-H3 structure (template). The SAAB + pipeline also performs structural clustering of CDR-H3 templates. To find structural templates with similar CDR-H3 loop shapes (analogous to canonical loop shapes), SAAB+ structurally clusters them based on their backbone atom RMSD values (see Methods). SAAB + outputs a tab delimited text file that can easily be merged with the current AIRR-seq standard format file [23].

### Structural CDR-H3 coverage and template usage

We investigated the structures of CDR-H3s used across BCR repertoires of different B-cell types in the human and mouse data. Predicting approximate shapes of CDRs rather than creating refined three-dimensional antibody models allows the SAAB+ pipeline to structurally annotate BCR repertoires rapidly. SAAB+ can analyse ~4.5 million BCR sequences a day on a 40 core computing cluster (Intel Xeon E5-2699 v4 @ 2.20GHz). SAAB+ structurally annotated the human data within two days and the mouse data in 4–5 weeks. Table 1 shows the coverage achieved by FREAD for each species.

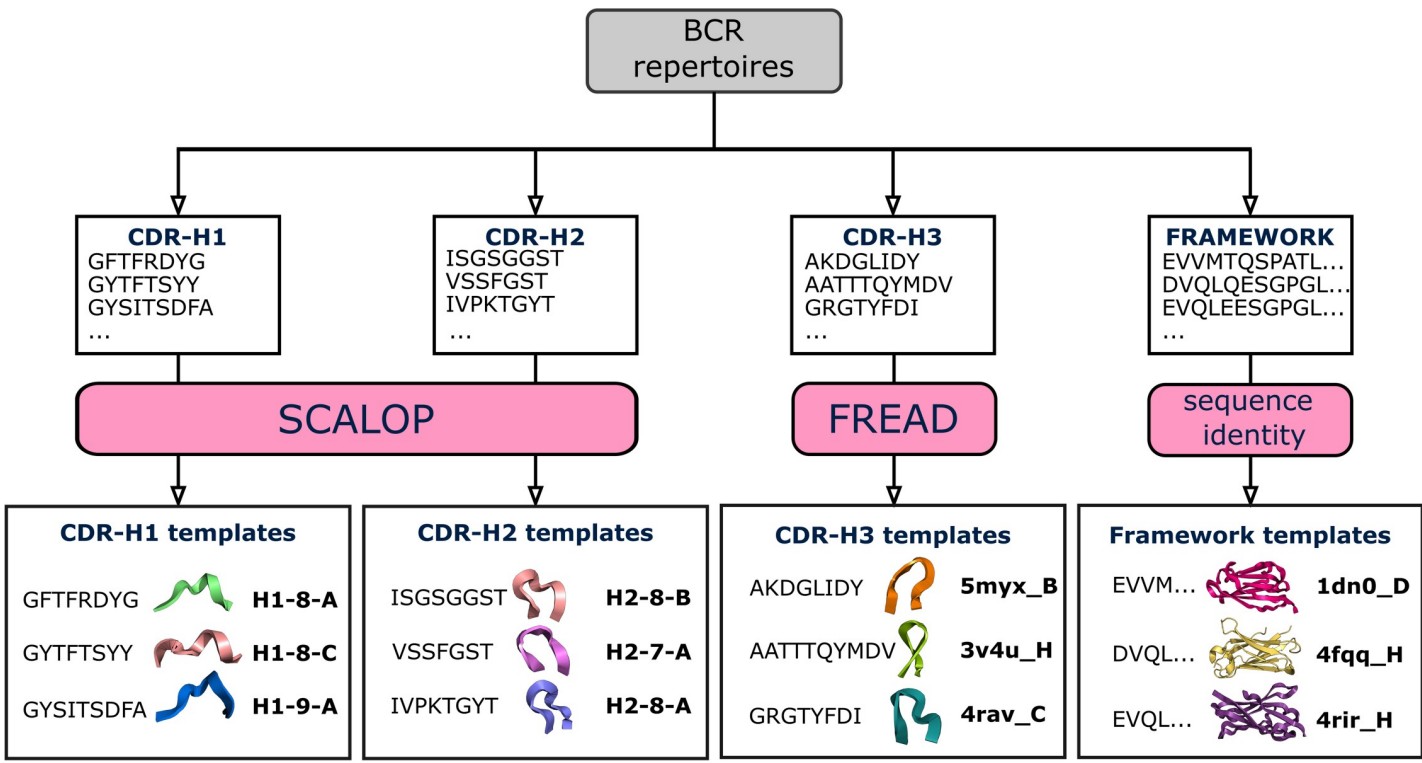

**Fig 1. Structural annotation of BCR repertoires.** BCR repertoires are sourced from the OAS resource. For each BCR sequence, CDR loop sequences are extracted, and the closest structural framework match is found, which is used in CDR-H3 loop grafting [15]. Next, SCALOP is used to identify canonical classes for non-CDR-H3 sequences, and FREAD is used to identify whether a CDR-H3 sequence shares a structure with any FREAD crystallographically-solved structures (templates). SCALOP returns a canonical class cluster identification (e.g. H1-8-A); FREAD returns the PDB code of an antibody structure with a protein chain specified (e.g. 5myx_B) [18], a CDR-H3 structural template.

CDR-H3 structural coverages of BCR repertoires were similar across different B-cell types in the human data (Kruskal-Wallis test, p = 0.37), but varied in the mouse data (Kruskal-Wallis test, p< 0.001). In both species, the variance of coverage was lower in the BCR repertoires of antigen-unexperienced B-cells (S2 Fig). The mean structural coverage was higher for mouse CDR-H3s than for human CDR-H3s (Table 1). Differences in length distributions could be a major cause of this discrepancy, as CDR-H3 structures are harder to predict for longer lengths, and the most common lengths were 11 and 12 residues in the mouse data, compared to 15 residues in the human data (S3 Fig).

Human and mouse BCR repertoires are the effector products of two different sets of germline genes. We therefore investigated whether species germline genes might also translate into preferred CDR-H3 structure usage. We used reported species origin information from SAb-Dab [24] to calculate the usages of different species CDR-H3 templates across our BCR repertoires (S4 Fig). As expected, human and mouse data used different frequencies of species

**Table 1. FREAD coverage of Ig-seq data.** The human data contained 5.7 million sequences with CDR-H3 loop lengths of 16 amino acids or shorter (see Methods). FREAD generated predictions for 48.1% of CDR-H3s in the human data, with an average coverage of 47.2% across BCR repertoires. The total number of mouse sequences was ~207 million, of which 88% were structurally-annotated with FREAD. The average structural coverage across mouse BCR repertoires was 88.1%.

| Data | Total sequences | CDR-H3 template predicted | Mean coverage with std |
|---|---|---|---|
| Human | 5,712,939 | 2,750,469 (48.1%) | 47.2±11% |
| Mouse | 206,680,496 | 182,309,575 (88%) | 88.1±4% |

CDR-H3 templates. The human BCR repertoires tended to use more human CDR-H3 templates as compared to uniform CDR-H3 template sampling, with mouse CDR-H3 templates appearing about as often as would be expected at random. In the mouse data, usage of mouse CDR-H3 templates was enriched, whilst usage of human CDR-H3 templates was reduced. These usages were roughly similar across B-cell types in both human and mouse data, suggesting a species bias towards CDR-H3 structural sampling largely independent of B-cell maturation. Interestingly, 109 (or ~4%) of all FREAD templates were never used in neither the human nor mouse data. Eighty eight of these templates were derived from nanobodies (S1 Appendix).

Together, these results suggest that different species may engage different epitopes on the same antigen through inherent structural biases.

## CDR-H3 cluster profiles along the B-cell differentiation axis

The adaptive immune system responds to antigen exposure by selecting and optimizing the most efficacious BCRs. Therefore, B-cells at different maturation stages may possess discrete paratope structural properties.

Galson et al., [7] demonstrated that different B-cell types could be separated using three heterogeneous sequence descriptors (clonality, average CDR-H3 loop length and percentage of V gene mutations) in a principal component analysis (PCA). We repeated their experiment on our human and mouse data (Fig 2A and 2B). In the human data, their sequence descriptors distinguished B-cell types. In the mouse data, pre, naïve, and plasma IGHM BCR repertoires clustered together, whilst plasma IGHG were clearly distinguishable from other B-cell types.

We investigated whether the structural annotation of CDR-H3s on its own could distinguish the BCR repertoires of different B-cell types, by performing PCA on CDR-H3 cluster usages across BCR repertoires. We found a clear separation of B-cell types in both the human and mouse data (Fig 2C and 2D), with a sequential pattern of B-cell differentiation in the human data (Naïve → Marginal → Memory → Plasma). Mouse IGHM and IGHG plasma BCR repertoires can be distinguished by CDR-H3 cluster usages, whereas neither we nor Galson et al., [7] observe the same separation in the human plasma BCR repertoires. The variance of CDR-H3 cluster usages in plasma IGHM were, in fact, more similar to antigen-unexperienced than to plasma IGHG BCR repertoires in the mouse data. Inaccuracies arising during B-cell sorting could cause improper B-cell labelling, adding noise to the B-cell type separation seen in Fig 2. In laboratory mice, the range and degree of antigen exposure is limited by pathogen-free housing conditions and low organism ages. This "purity" could account for the finer separation of B-cell types.

To quantify the behaviour seen in Fig 2, we employed the DBSCAN clustering algorithm with increasing maximum distance to closest neighbours (ε) to interrogate the densities of CDR-H3 cluster usages across BCR repertoires. Clustering at lower ε distances indicates a more similar distribution of CDR-H3 cluster usages. In the human data, all naïve BCR repertoires clustered at low ε distances along with one marginal zone BCR repertoire. As the value of ε was increased, all marginal zone BCR repertoires merged with the naïve BCR repertoire cluster, followed by memory and finally plasma BCR repertoires (S5 Fig). In the mouse data, pre and naïve BCR repertoires initially formed two separate clusters at low ε distances. As ε was increased, antigen-unexperienced (pre and naïve) BCR repertoire merged into a single cluster, followed by plasma IGHM and plasma IGHG repertoires respectively (S6 Fig).

BCR repertoires of different B-cell types are known to have their own characteristic distributions of CDR-H3 lengths [7,25]. To see whether this alone was driving the separation, we repeated our PCA experiment at specific lengths of CDR-H3, again employing DBSCAN to

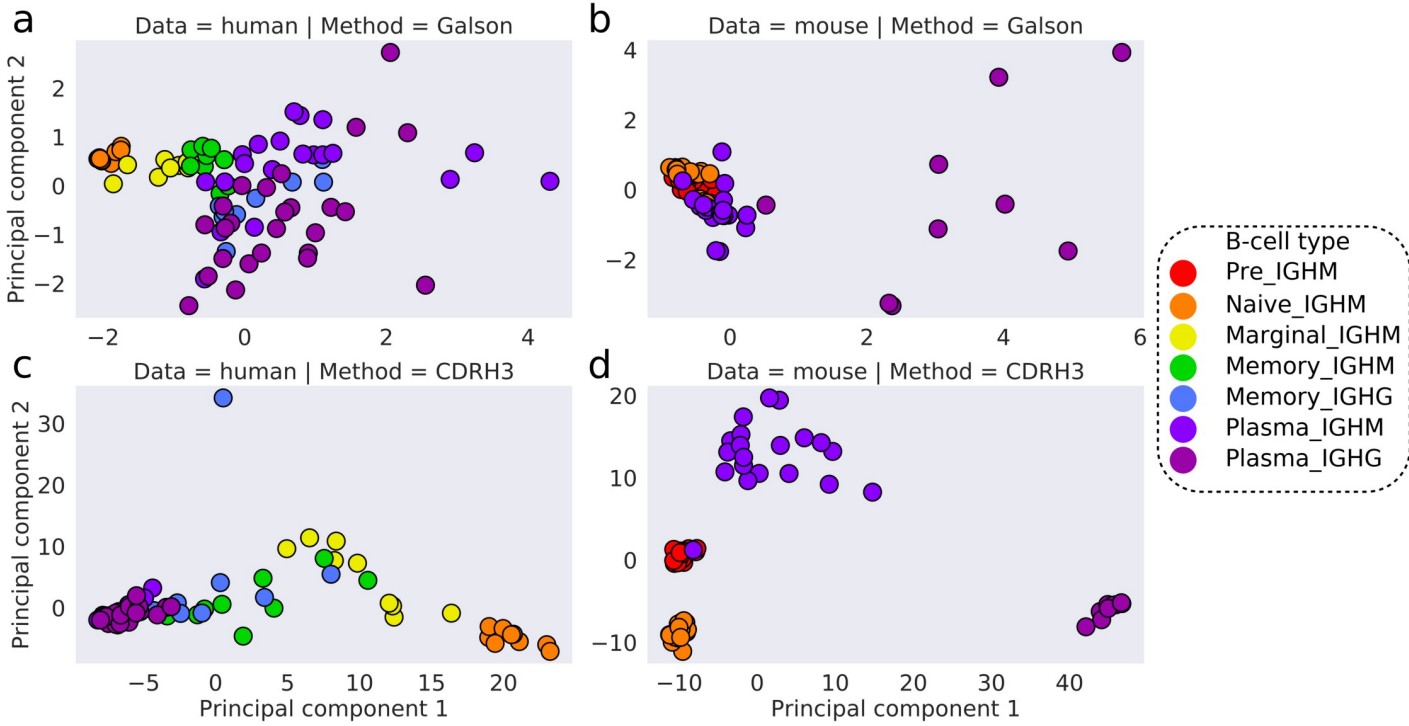

**Fig 2. PCA on the human and mouse data.** Features included in the PCA were either average CDR-H3 length, clonality and percentage of SHMs in V genes (**a, b**) or CDR-H3 cluster usages (**c, d**). The human data is shown in **a** and **c**, whilst the mouse data is in **b** and **d**. The first two principal components are used to visualize the separation of BCR repertoires. Colours represent different B-cell types.

interrogate the densities of CDR-H3 cluster usages. For each length, we observed the same patterns, confirming that our separation of BCR repertoires was not solely an artefact of CDR-H3 loop length (S7 Fig).

These findings give structural confirmation to our understanding of B-cell development from antigen-unexperienced to terminally-differentiated plasma B-cells. The collection of CDR-H3s in a terminally-differentiated BCR repertoire should be reflective of individual's complex history of antigenic stimulations yielding highly specialized, high-affinity antibodies [2]. These results demonstrate a mode of structural BCR repertoire ontogeny, where antigen-unexperienced BCR repertoires have the most conserved "public" frequencies of CDR-H3 structural clusters across individuals. Upon antigenic stimulation, the somatic hypermutation (SHM) machinery of B-cells recursively introduces point mutations, primarily to the antibody CDR regions [3,26]. Our DBSCAN analysis shows that BCR repertoires of different B-cell types do not use equal frequencies of CDR-H3 clusters, suggesting that affinity maturation leads to discernible structural changes in the paratope. As B-cells differentiate to the next developmental stage, their repertoires become more personalized; a fine-tuning of antibody CDR structures along the differentiation axis.

Next, we checked whether above results were caused by varying numbers of utilized CDR-H3 clusters. We evaluated the total number of CDR-H3 clusters represented across different B-cell types in the human and mouse data (Fig 3A and 3B). None of the BCR repertoires used the maximum number of CDR-H3 clusters (1,169), and the numbers varied between BCR repertoires, with antigen-unexperienced repertoires using the most. The average number of CDR-H3 clusters in plasma IGHG BCR repertoires was 3–4 times smaller than in naïve repertoires.

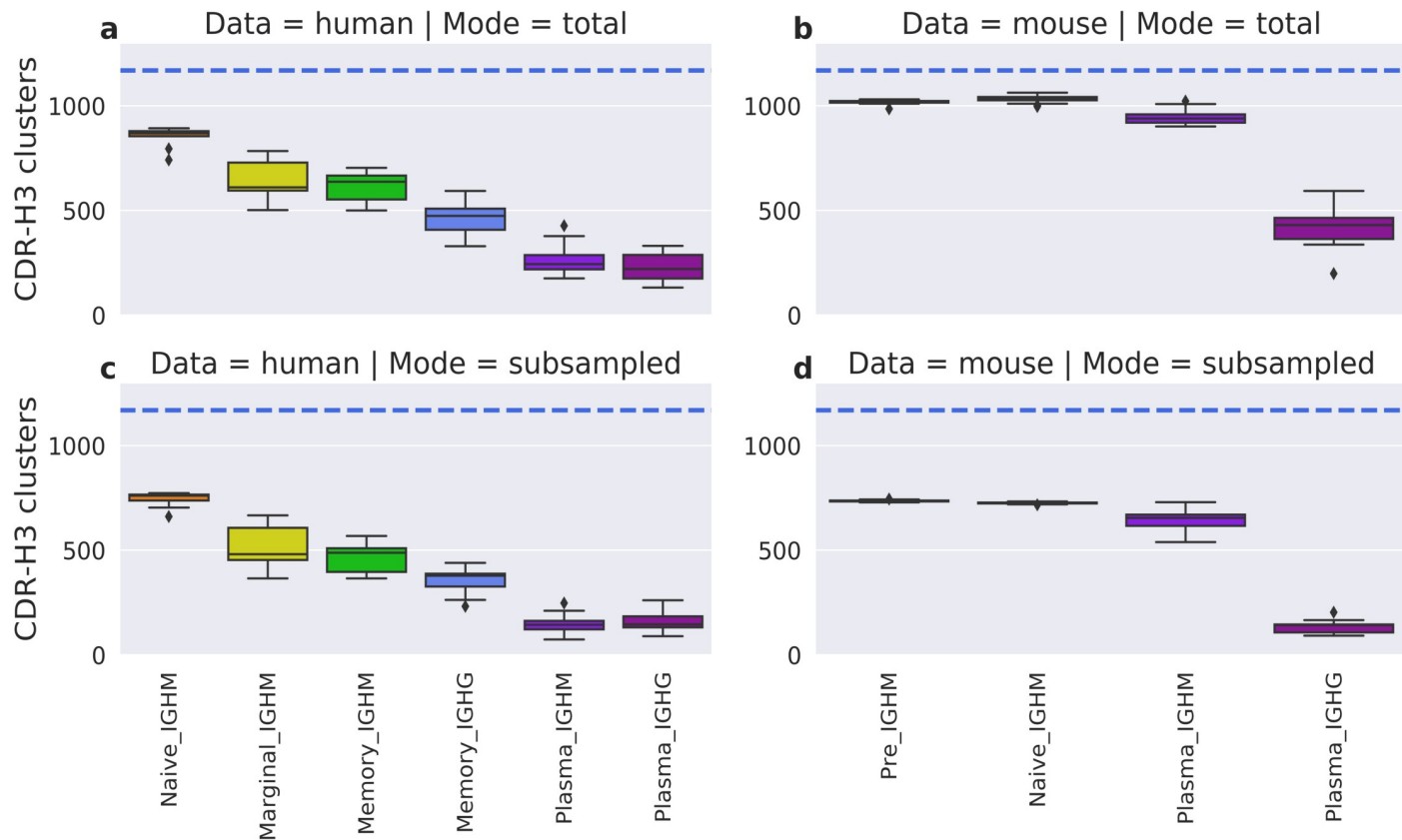

**Fig 3. Average number of CDR-H3 clusters in the human and mouse data.** The top boxplots depict the total number of CDR-H3 clusters in human (**a**) and mouse (**b**) BCR repertoires. In the bottom boxplots, every human (**c**) and mouse (**d**) BCR repertoire was subsampled 100 times for 10,000 sequences, with the average number of CDR-H3 clusters recorded. Colours represent different B-cell types. The horizontal blue line shows the total number of CDR-H3 clusters in our FREAD library, and therefore the theoretical maximum.

This difference could potentially be explained by a smaller number of isolated plasma B-cells. To account for the varying sizes of BCR repertoires, we subsampled 10,000 sequences from each of them 100 times and recorded the average number of CDR-H3 clusters. The subsampling gave a similar pattern to the complete data, with the average number of CDR-H3 clusters being highest in antigen-unexperienced BCR repertoires (Fig 3C and 3D), and total numbers of represented clusters decreasing along the B-cell differentiation axis. This drop in the number of CDR-H3 clusters is not caused by poorer structural coverage of more differentiated BCR repertoires, as we have already shown that the coverage is not significantly different across B-cell types in the human data, and increases for more differentiated cells in the mouse data (S2 Fig). Therefore, we suspect that this decrease in the number of represented CDR-H3 clusters along the differentiation axis was the result of only specific CDR-H3 structures transitioning to the next development stage.

To confirm this hypothesis, we investigated whether the decreased numbers of CDR-H3 clusters in antigen-experienced BCR repertoires are also accompanied by structural specialization i.e. personalized CDR-H3 cluster usage. We employed Shannon entropy to investigate the structural diversity of CDR-H3s across our BCR repertoires. High entropy demonstrates a high diversity of CDR-H3 structures, whilst low entropy indicates the over-representation of one or more CDR-H3s. To account for the decreasing number of represented CDR-H3 structures, we calculated the proportion of theoretical maximum entropy for each BCR repertoire

to yield a normalized estimate of the diversity of CDR-H3 clusters used (S8 Fig). This confirmed that the structural diversity of CDR-H3 gradually decreased along the B-cell differentiation axis. Antigen-unexperienced BCR repertoires had the highest structural diversity of CDR-H3s, as well as the lowest variance in entropy across B-cell types. Marginal and memory IGHM BCR repertoires utilized the same number of CDR-H3 structures (p = 0.66, Mann-Whitney U-Test), whilst the structural diversity was significantly lower in memory B-cells (p = 0.005, Mann-Whitney U-Test). Our results again give structural confirmation of the affinity maturation process, where only CDR structures that are specific to cognate antigens are retained.

Overall, the above results demonstrate that B-cell types can be distinguished based on the profile of CDR-H3 structural descriptors alone and that antigen-unexperienced BCR repertoires utilized the highest number and the highest entropy of CDR-H3 clusters. Cluster frequencies in naive BCR repertoires were conserved across different B-cell donors. As B-cells differentiate, their CDR-H3 cluster usage becomes narrower and more distinct between individuals, which is reflective of both affinity maturation and a personalized history of B-cell selection. These results provide us with the first structural insight into fundamental processes that govern BCR repertoire differentiation across B-cell donors.

## Canonical class characterization

Our analysis so far has focused on CDR-H3, but CDR-H1 and CDR-H2 also play a key role in shaping the antibody paratope [27]. Most CDR-H1 and CDR-H2 loops are found in a small set of structures known as canonical classes. This allows prediction of their structure from sequence with high confidence [21].

A single V gene encodes for both CDR-H1 and CDR-H2 loops and it is known that SHMs preferentially take place in these loops during B-cell differentiation [3,26]. As the level of SHMs increases with B-cells differentiation, the number of mutations in the V gene has often been used as a proxy to study B-cell development [7,28].

Here, we investigated whether SHMs in the V gene lead to structural changes in CDR-H1 and CDR-H2 in humans and mice. We calculated the percentage of sequences across BCR repertoires where either the CDR-H1 or CDR-H2 canonical class diverged from its parent germline. Sequences with unassigned canonical class information were retained in the analysis as their number was low (S2 Table), and SHMs can still change loop conformation to a yet uncharacterized canonical class. As of June 2019, only one human and six mouse V genes contained either a CDR-H1 or a CDR-H2 shape that did not fall into a SCALOP canonical classes [21].

Canonical class divergence from germline occurred in all B-cell types, but was observed to increase along the B-cell differentiation axis in the human data (S9 Fig). This was less clear in the mouse data. Pre and naïve B-cells had less canonical class divergence from the germline, whereas memory and plasma B-cells had a higher divergence. These results place structural information on the knowledge that the percentage of V gene mutations increases with B-cell differentiation [7]. The average percentage of canonical class divergence across B-cell types were consistently higher in human than mouse data. This is in agreement with previously-reported results showing that human V genes tend to accumulate a larger number of SHMs than mouse [29].

CDR-H1 and CDR-H2 loops had different levels of canonical class divergence in both human and mouse data, with CDR-H1s changing their germline loop shapes more often than CDR-H2s (S10 Fig). This can probably be directly attributed to the different number of canonical classes accessible to CDR-H1 and CDR-H2 (7 versus 4), which implies CDR-H1 loops have a greater degree of structural freedom.

Both Galson et al., [7] and Greiff et al., [9] studies showed that the V gene usages varied across B-cell types. Here, we investigated whether canonical class usages could provide a structural explanation for the observed alterations in V gene utilization during B-cell differentiation. As with CDR-H3, we performed PCA on combinations of canonical class usages across BCR repertoires (S11 Fig). In the human data, we found a separation between naïve and more differentiated B-cell types, with naïve BCR repertoires utilizing more similar canonical class usages. In the mouse data, BCR repertoires were separated into different B-cell types with the sequential pattern of B-cell differentiation.

Our results demonstrate that canonical class usages are not static during B-cell differentiation, with more mature B-cells exhibiting a higher level of canonical class divergence from the parent germline. CDR-H1 and CDR-H2 structures are clearly modulated to help refine the antibody paratope configuration against the cognate antigen.

## Patterns of CDR-H3 cluster usage

Biased usage of CDR-H3 clusters is observed in different BCR repertoires along the differentiation axis. For instance, antigen-unexperienced B-cells share the closest frequencies of CDR-H3 clusters (Fig 2). A detailed understanding of biased CDR-H3 structure usage would significantly advance our knowledge of the adaptive immune system development and maturation.

To investigate patterns of biased CDR-H3 cluster usage, we split CDR-H3 clusters into three groups for each B-cell type based on frequencies of CDR-H3 clusters used across these BCR repertoires. "Structural Stems", which were defined as CDR-H3 clusters, whose frequencies were significantly over-represented across the BCR repertoires of a given B-cell type, "Under-Represented" which describes under-represented CDR-H3 clusters. And CDR-H3 clusters, whose frequencies were not significantly different from random uniform sampling -"Random-Usage" (Fig 4).

First, we looked at the average number of CDR-H3 clusters found in our three groups (Structural Stems, Random-Usage and Under-Represented) across the different B-cell types. In all BCR repertoires, Under-Represented always contained the largest number of CDR-H3 clusters (S12 Fig), however, this does not translate to dominance in terms of coverage (Fig 5). This is because, in most cases, Under-Represented CDR-H3 clusters tend to have only a few sequences in a repertoire that share that shape, whereas Structural Stems will have far higher numbers.

In the human data, the number of Structural Stems was largest in naïve BCR repertoires and gradually decreased along the B-cell differentiation axis. The number of Random-Usage CDR-H3 clusters was lowest in the naïve repertoires. This number increased in marginal BCR repertoires followed by a gradual decline along the B-cell differentiation axis. Similar to the human data, the number of Structural Stems was the highest in antigen-unexperienced BCR repertoires in the mouse data. The number of Structural Stems declined in plasma IGHM and were completely absent in plasma IGHG repertoires.

Next, we investigated the proportional composition of BCR repertoires across B-cell types with Structural Stem, Random-Usage and Under-Represented CDR-H3 clusters. The distribution of repertoire coverages differed between B-cell types in both human and mouse data (Fig 5). Structural Stems cover ~70–80% of antigen-unexperienced BCR repertoires, with coverage declining along the B-cell differentiation axis. In contrast, coverage with Under-Represented clusters gradually increased as B-cells matured. Pre and naïve BCR repertoires were least covered with Random-Usage CDR-H3 clusters (only 5–10%). In the human data, coverage with Random-Usage CDR-H3 clusters showed a transient increase in memory BCR repertoires followed by a decline in plasma repertoires, though this trend was less evident in the mouse data.

## a CDR-H3 Clusters

- Cluster #1
- Cluster #2
- Cluster #3
- Cluster #4

## b Observed Naïve BCR Repertoires

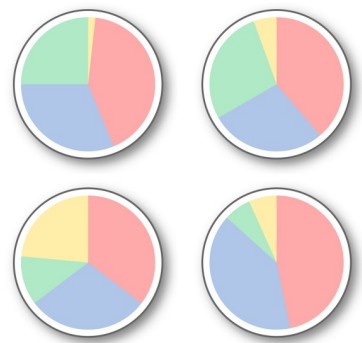

## c Randomly Sampled (RS) Naïve BCR Repertoires

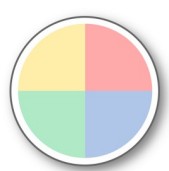

## d Cluster Usage

| Cluster ID | RS | Observed | | | | Classification |
|---|---|---|---|---|---|---|
| #1 | 25 | 41 | 33 | 30 | 45 | Structural Stem |
| #2 | 25 | 30 | 29 | 30 | 37 | Structural Stem |
| #3 | 25 | 27 | 27 | 16 | 8 | Random-Usage |
| #4 | 25 | 2 | 11 | 24 | 10 | Under-Represented |

**Fig 4. Pattern of CDR-H3 cluster usage within a specific B-cell type.** A schematic representation of how we grouped CDR-H3 clusters based on their pattern of usage. (**a**) In this mock example, only four CDR-H3 clusters are found in (**b**) four naïve BCR repertoires. (**c**) In the case of random uniform sampling, each of these clusters would constitute approximately 25% of a simulated BCR repertoire. (**d**) Structural Stems are defined as CDR-H3 clusters, which are over-represented across BCR repertoires when compared to random cluster usage. Under-represented are clusters that are under-represented across repertoires. CDR-H3 clusters, which usages are not significantly different from random sampling, were termed Random-Usage.

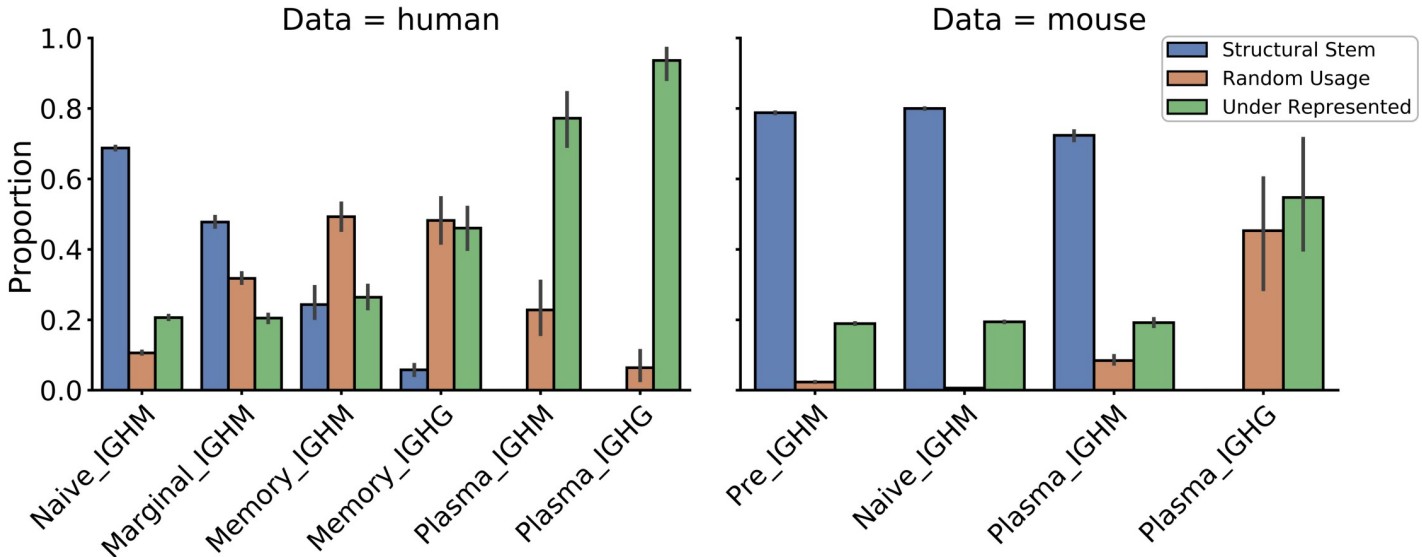

**Fig 5. Coverage of BCR repertoires with CDR-H3 clusters based on their pattern of usage in the human and mouse data.** The X-axis shows different B-cell types in the order of the B-cell differentiation axis. The Y-axis shows the proportion coverage of BCR repertoire sequences with CDR-H3 clusters.

The same CDR-H3 clusters are preferentially over-represented across different B-cell types with the number of these over-represented CDR-H3 clusters diminishing to none along the B-cell development axis (S13 Fig) (S2 Appendix).

These results demonstrate that antigen-unexperienced BCR repertoires display CDR-H3 structural conservatism. Naive BCR repertoires are largely composed of "public" sets of over-represented CDR-H3 clusters. During B-cell selection, CDR-H3 cluster usages become less conserved across individuals as the coverage with Random-Usage and Under-Represented CDR-H3 clusters rise. In terminally-matured plasma IGHG BCR repertoires, none of CDR-H3 clusters was significantly over-represented across individuals. This reflects how the history of antigenic stimulations structurally shapes BCR repertoires, which become increasingly specialized as B-cells differentiate.

## Discussion

We have carried out the first systematic study of structural diversity in the BCR repertoires of multiple donors and species along the B-cell differentiation axis. By mapping sequences to solved antibody structures, we show the structural transformation occurring as BCR repertoires develop in humans and mice.

Our data show that B-cell types can be distinguished based solely on the structural diversity of CDR-H3 loops. Antigen-unexperienced (pre and naïve) BCR repertoires contain conserved "public" CDR-H3 cluster frequencies across individuals. As B-cells differentiate, their structural repertoires become increasingly personalized, as a reflection of each individual's history of antigen exposure. Antigenic stimulation induces marked changes in the pattern of CDR-H3 cluster usage in BCR repertoires. The repertoires utilize a smaller number of available CDR-H3 configurations, CDR-H3 structural diversity is reduced, and CDR-H3 cluster usage becomes increasingly divergent from naïve BCR repertoires. Structural changes also take place in non-CDR-H3 loops, highlighting the importance of canonical loops in paratope shaping. This shows how structure changes as B-cells, whose CDRs are complementary to cognate antigens, are positively selected.

The human and mouse data used in this work came from two studies that interrogated BCR repertoire sequence diversities following antigenic stimulation [7,9]. In their original publications, the sequence diversity was shown to decrease along the B-cell differentiation axis, with the most marked decline in the plasma cell repertoires in response to antigen exposure [7,9]. In our work, we find evidence to support these diversity changes on the structural level, which can be indicative of affinity selection of antigen specific BCRs. The SAAB+ pipeline was able to capture both these sequence/structural convergence paradigms, where similar CDR-H3 templates are more likely to be selected in less sequence diverse BCR repertoires as well as identification of sequence dissimilar CDR-H3 loops located in the same structural space.

Most publicly available Ig-seq studies interrogate BCR repertoires either in disease or antigenic stimulation settings [17]. This could potentially lead to sequence/structural diversity measurements that are different from the resting BCR repertoire state. Recently, two Ig-seq studies released cumulatively more than 7 billion BCR reads of unsorted B-cells from healthy individuals [2,30]. These studies provided crucial insights into the BCR sequence diversity and dynamics amongst healthy individuals. However, analogous large-scale studies have not yet been performed on sorted BCR repertoires due to high costs concomitant with experimental setups, and labour requirements. In order to develop a robust immunodiagnostics pipeline, it is important to understand sequence/structural diversity on the level of the whole repertoire as well as individual B-cell types.

In this work we considered only the three CDRs encoded by heavy chain genes. However, the light chain also plays an important role in shaping the BCR paratope [31]. Therefore, it is anticipated that the diversity of light chain CDR configurations would also change along the B-cell differentiation axis. To advance our understanding of the role of the light chain in paratope shaping, further investigations are required to study joined structural diversities of heavy and light chains within individual B-cell donors.

Inclusion of cognate light chain pairing information into our analysis would also facilitate generation of refined antibody models. Increased availability of paired heavy/light BCR data [32] and improvements in antibody modelling speed [13] will pave the way to a new frontier of antibody structure usage analysis at the scale of an entire BCR repertoire. Structural descriptors harvested from these models will push forward the resolution of our current work, enabling calculations of paratope charge and hydrophobicity, as well as antibody developability profiles [33].

Structural coverage of CDR-H3s of the human and mouse data was $\sim 48\%$ and $\sim 88\%$ respectively. In this analysis, we did not consider BCR sequences with CDR-H3 loop lengths greater than 16 amino acids to ensure high prediction accuracies. This modelling quality filter removed 3.9% ($\pm$ 2 s.t.d) of the mouse and 29% ($\pm$11 s.t.d) of the human BCR datasets from the analysis, meaning that the structural diversity information of longer CDR-H3 loops unexplored (S3 Fig). Despite these marked differences in BCR repertoire coverages, both human and mouse data showed similar patterns of structural diversities along the differentiation axis. These findings agree with and provide structural reasons for the sequence diversity measurements calculated across all lengths in the original studies of the human and mouse data. Therefore, it is unlikely that the unexplored portions of human BCR repertoires would have significantly different structural diversity properties.

More than 400 antibody structures have been solved in 2019 [24], which constitutes more than 10% of our CDR-H3 template library. With this steadily increasing rate of antibody structure availability and continuous improvements in homology modelling technologies, further studies will be soon necessary to investigate trade-offs between BCR repertoire coverage and prediction accuracies.

Structural characterization of Ig-seq data can augment existing analysis pipelines [13]. Current Ig-seq data clustering approaches work on the premise that CDR-H3 sequence identity alone can capture structural features of the paratope [6]. However, sequences with low CDR-H3 sequence identity can adopt close shapes and *vice versa* [13]. Hence, the development of structure-aware clustering methods such as SAAB+ allows for the direct grouping of structurally/functionally related BCR sequences [34], as well as enables structural changes to be traced within individual B-cell linages.

A set of CDR-H3 clusters was consistently over-represented across all B-cell donors ("Structural Stems") within the specific B-cell types. These clusters encompassed 70–80% of all sequences in antigen-unexperienced BCR repertoires. This shows that humans and mice largely rely on a conserved "public" set of CDR-H3 clusters to initiate antigen recognition. This knowledge could be leveraged to study immune system disorders, including immunosenescence, where distortions in the conserved public pattern of CDR-H3 cluster usage in antigen-unexperienced BCR repertoires could signal disease states. Furthermore, the knowledge of over-represented CDR-H3 clusters in naïve BCR repertoires could be applied in rational phage display library engineering, with Structural Stem cluster sequences used as starting points for library diversity generation.

Recently, transgenic mouse models with human adaptive immune system have been created to raise "naturally human" antibodies in non-human systems [35]. However, their BCR repertoires are shaped inside the murine environment, which could potentially select for BCR

paratopes non-native to the human body. Hence, our structural diversity analysis could also be employed in the paratope "humanness" assessment of BCR repertoires derived from transgenic animals.

## Methods

### Data selection

Human Ig-seq data from Galson et al., [7] and mouse (C57BL/6 inbred strain) Ig-seq data from Greiff et al., [9] were used. Galson et al., ("human") is a longitudinal vaccination study across nine healthy human donors, in which the heavy chain of naïve, marginal zone, memory, and plasma B-cell types were interrogated [7]. Greiff et al., ("mouse") is a high depth sequencing study of the murine adaptive immune system in response to antigenic stimulation, containing heavy chain BCR repertoires from pre, naïve and plasma B-cells [36]. Both studies used FACS to sort B-cells into subpopulations according to their differentiation stages.

The Ig-seq amino acid sequences were downloaded from the Observed Antibody Space (OAS) [17] resource, retaining their Data Unit information. Each Data Unit is a sequencing sample from a single B-cell donor with a defined combination of B-cell type and isotype information, and contains sequences that are IMGT-numbered [19] and filtered for antibody structural viability. Henceforth, OAS Data Units will be referred to as B-cell receptor (BCR) repertoires.

To investigate structural changes along the B-cell differentiation axis, BCR repertoires with defined B-cell type and isotype information were downloaded. Only IGHG and IGHM sequences were considered as these were the most abundant. The total number of BCR repertoires in the human and mouse data were 85 and 82 respectively.

### Structural annotation

To annotate the human and mouse data with structural information, we developed a customized version of our SAAB pipeline [15], SAAB+ that predicts the structural shape of the IMGT-defined CDRs. CDR-H1 and CDR-H2 adopt a limited number of structural configurations, known as canonical classes [16,37,38], which can be predicted accurately and rapidly from sequence [21]. SAAB+ uses SCALOP [21] to annotate non-CDR-H3 loop canonical classes. Canonical class annotation should be highly accurate, with SCALOP predictions estimated to be within 1.5 Å of the true structure 90% of the time [21]. The June 2019 SCALOP database was used in this study.

SAAB+ uses FREAD to predict CDR-H3 structural templates [22,39,40]. Accurately modelling all the CDR-H3s in an Ig-seq dataset is challenging, owing to the vastness of structural space accessible to these loops [41–43], relative to the small number of publicly-available crystallographically-solved antibodies (many of which are highly sequence redundant) [24]. In addition, structurally-solved antibodies have a CDR-H3 length distribution and sequence diversity that is different from natural Ig-seq data (S3 Fig). We tested the performance of FREAD on the Ig-seq data and, at the parameters used, the expected average RMSD of FREAD CDR-H3 template predictions for both human and mouse data is 2.5 Å (see S3 Appendix). This is in line with current state-of-the-art CDR-H3 modelling software tools (mean RMSD of 2.8 Å) [44]. In a similar manner to DeKosky et al., [14], we limited our CDR-H3 analysis to loop lengths of 16 amino acids or shorter, as far fewer structures with longer CDR-H3 loops are available and longer loops have increased structural freedom. We also excluded CDR-H3 loops shorter than five amino acids from our analysis, as only three CDR-H3 templates covered these lengths. FREAD templates were downloaded from SAbDab (14[th] November 2018)

[24], and consisted of all X-ray crystal structures of antibodies with a resolution better than 2.9 Å.

### CDR-H3 clustering

To identify similar CDR-H3 loop structures, we used the DTW algorithm [16] to cluster FREAD templates by backbone RMSD. Those within 0.6 Å were placed in the same cluster, reducing our 2,943 FREAD CDR-H3 templates to 1,169 CDR-H3 clusters.

### Filtering BCR repertoires

As PCR sequencing can lead to variable amplicon amplification, we removed any BCR repertoire if its two most redundant CDR-H3 clusters contained more than 80% of all repertoire sequences (S1 Fig). We also discarded any BCR repertoire that contained fewer than 10,000 sequences with predicted CDR-H3 structures—this cut-off was selected to allow for adequate sampling of CDR-H3 template usages, whilst retaining as many BCR repertoires as possible (S3 Table). This reduced the number of repertoires for all subsequent structural analysis to 81 (human) and 73 (mouse). CDR-H1 and CDR-H2 loops were not taken into account in determining BCR repertoire quality, since canonical class coverage was ~95% and ~99% for the human and mouse data respectively (S2 Table).

### Patterns of CDR-H3 cluster usage

We analysed the pattern of CDR-H3 cluster frequencies in the human and mouse data, to identify clusters whose usages were over-represented (Structural Stems), random (Randomly-Used) and under-represented (Under-Represented) within a given B-cell type.

The structurally-annotated human and mouse data was split into individual groups based on unique B-cell type and isotype combinations. Within these groups, we calculated the CDR-H3 length distributions and the proportion modellable by FREAD for each CDR-H3 length. Next, we randomly selected CDR-H3 templates from our FREAD library (with replacement) according to these distributions, to generate a randomized dataset for each BCR repertoire. Sampling was performed across the set of FREAD templates already present in each BCR repertoire. The randomized dataset sizes were set to one million sequences and the total number of randomized datasets was matched to the number of the BCR repertoires within the corresponding groups (S3 Table).

A one-sided Mann-Whitney rank test (p = 0.05) was performed on the relative usage of each CDR-H3 cluster in the grouped BCR repertoires and the corresponding randomized datasets, to categorize them as Structural Stem, Random-Usage or Under-Represented CDR-H3 clusters.

### Statistical analysis

Statistical analyses were performed in Python using the scikit-learn[45] and scipy packages. Detailed information on statistical tests is outlined in the figure legends. Data visualization was performed with the seaborn package.

## Supporting information

**S1 Fig. Percentage of sequences contained within the two most redundant CDR-H3 clusters for each BCR repertoire in the human and mouse data.** Each circle represents the percentage of sequences present in the two most redundant CDR-H3 clusters in a BCR repertoire. If this percentage exceeds 80% of the total number of that BCR repertoire's sequences (orange

circle), the BCR repertoire was not included in the subsequent structural analysis. Different thresholds were checked, lower percentage cut-offs produced the same qualitative results in our structural analysis. Hence, 80% cut-off was selected to retain as many BCR repertories as possible.
(TIF)

**S2 Fig. CDR-H3 structural coverage in human (A) and mouse (B) BCR repertoires.** FREAD CDR-H3 modellability was calculated across BCR repertoires of different B-cell types. FREAD modellability is defined as the percentage of sequences with predicted CDR-H3 structures over the total number of sequences in a given BCR repertoire.
(TIF)

**S3 Fig. Normalized CDR-H3 length distribution in the human and mouse data, and our FREAD CDR-H3 template library.** Normalized CDR-H3 length distribution was calculated for all BCR repertoires in the human and mouse data, and all CDR-H3 templates that were in our FREAD library. Mouse CDR-H3s were on average shorter than human. The distribution of FREAD CDR-H3 lengths is different that found in the human and mouse data. The vertical blue line shows our CDR-H3 length cut-off (17 residues). Sequences and FREAD templates whose CDR-H3 lengths were longer than the cut-off were not considered in our analysis.
(TIF)

**S4 Fig. Reported species origin of CDR-H3 templates in the human and mouse data.** We calculated the proportion of CDR-H3 template reported species of origin for every BCR repertoire across different B-cell types. Reported species origin information was extracted from SAbDab (10). Orange bars show the proportion of human CDR-H3 templates, blue bars represent mouse CDR-H3 templates, while cyan bars depict the proportion of other than human or mouse CDR-H3 templates. The horizontal lines represent the expected outcome of uniform sampling of human (orange) and mouse (blue) CDR-H3 templates. Uniform sampling was calculated as the number of human or mouse CDR-H3 templates over the total number CDR-H3 templates found in our FREAD library.
(TIF)

**S5 Fig. Analysis of densities of CDR-H3 cluster usage in the human data.** (**a**) PCA was performed on the human BCR repertoires (circles) with CDR-H3 cluster usages used as the features. The first two principal components were used to visualize any separation. Colours represent different B-cell types. (**b**) DBSCAN analysis with increasing maximum distance (ε) was employed to interrogate CDR-H3 cluster usage densities across human BCR repertoires. PCA analysis (as in **a**) was then used to visualize the DBSCAN clustering. The parameter ε was increased left-to-right, top-to-bottom. Marker shapes indicate different B-cell types; blue colour represents BCR repertoires that clustered with antigen-unexperienced BCR repertoires (naïve); orange colour shows DBSCAN-unclustered BCR repertoires at that ε value.
(TIF)

**S6 Fig. Analysis of densities of CDR-H3 cluster usage in the mouse data.** (**a**) PCA was performed on the mouse BCR repertoires (circles), with CDR-H3 cluster usages used as the features. The first two principal components were used to visualize any separation. Colours represent different B-cell types. (**b**) DBSCAN analysis with increasing maximum distance (ε) was employed to interrogate CDR-H3 usage densities across mouse BCR repertoires. PCA analysis (as in **a**) was then used to visualize the DBSCAN clustering. The parameter ε was increased left-to-right, top-to-bottom. Marker shapes indicate different B-cell types; cyan colour (in the top left subplot) represents naïve BCR repertoires, blue colour represents BCR

repertoires that clustered with antigen-unexperienced BCR repertoires (pre and naïve); orange colour shows DBSCAN-unclustered BCR repertoires at that ε value.
(TIF)

**S7 Fig. Structural interrogation of the human and mouse data with specific CDR-H3 lengths.** PCA was performed on (**a**) human and (**b**) mouse BCR repertoires, with CDR-H3 cluster usages selected as the features. The first two principal components were used to visualize any separation. Colours represent different B-cell types. DBSCAN was employed to quantify densities of CDR-H3 cluster usages across the repertoires. Marker shapes illustrate DBSCAN cluster information. Circle markers indicate DBSCAN unclustered BCR repertoires; other marker shapes show individual DBSCAN clusters. We considered a B-cell type separation if naïve and pre (antigen-unexperienced) BCR repertoires displayed the closest densities of CDR-H3 cluster usages at lower ε values in DBSCAN. In both (**a**) human and (**b**) mouse repertoires, antigen-unexperienced B-cell types cluster first regardless of CDR-H3 lengths. This confirms that BCR repertoires of different B-cell types have different patterns of CDR-H3 cluster usage.
(TIF)

**S8 Fig. CDR-H3 structural diversity in the human and mouse data.** Shannon entropy was calculated for CDR-H3 cluster usage in human (**a**) and mouse (**b**) data. To account for the varying numbers of CDR-H3 clusters across B-cell types, structural diversity of CDR-H3s was expressed as a proportion of entropy over theoretical maximum entropy in the human (**c**) and mouse (**d**) data. Theoretical maximum entropy was found for each BCR repertoire by using CDR-H3 structures represented in the given repertoire in equal proportions in entropy calculations. Higher values indicate higher diversities of CDR-H3 cluster usage. The Mann-Whitney U-test was used for statistical analysis and p-values are reported.
(TIF)

**S9 Fig. Canonical class divergence from parent germline class in the human and mouse data.** Canonical class divergence was defined as a mismatch in either the CDR-H1 or CDR-H2 canonical class from the germline canonical class. Percentages were calculated as the number of sequences with canonical class divergence over the total number of sequences in a given BCR repertoire. Colours represent different B-cell types.
(TIF)

**S10 Fig. CDR-H1 and CDR-H2 canonical class divergence from parent germline class in the human and mouse data.** The top boxplots show canonical class divergence in CDR-H1 canonical class from the germline canonical class in the (**a**) human and (**b**) mouse data. Percentages were calculated as the number of sequences with canonical class divergence over the total number of sequences in a given BCR repertoire. The bottom boxplots show the same analysis on CDR-H2 canonical class in the (**c**) human and (**d**) mouse data. Colours represent different B-cell types.
(TIF)

**S11 Fig. PCA on the human and mouse BCR repertoires.** Features included in the PCA were frequencies of CDR-H1 and CDR-H2 combinations in BCR repertoires. The first two principal components were used to visualize the separation of BCR repertoires. Colours represent different B-cell types.
(TIF)

**S12 Fig. Number of CDR-H3 clusters based on their pattern of usage across different B-cell types in human and mouse data.** Structural Stems (blue bars) were defined as CDR-H3

clusters, which were over-represented across BCR repertoires of the same B-cell type. Under-Represented (green bars) were under-represented CDR-H3 clusters. CDR-H3 clusters, whose usages were not significantly different from random sampling, were termed Random-Usage (orange bars). The X-axis shows different B-cell types in the order of the B-cell maturation axis. The Y-axis shows the number of CDR-H3 clusters.
(TIF)

**S13 Fig. Overlap of Structural Stem CDR-H3 clusters between naïve and antigen experienced BCR repertoires in the human and mouse data.** Naïve and antigen experienced BCR repertoires were investigated for the Structural Stem overlap in the human (row **A**) and mouse (row **B**) data. The overlap was defined as a number of shared clusters between Structural Stem CDR-H3 clusters in BCR repertoires found in two different B-cell types. The X-axis shows the B-cell types. The Y-axis shows the total number of Structural Stem CDR-H3 clusters. The stripped pattern indicates the number of overlapped Structural Stem CDR-H3 clusters.
(TIF)

**S1 Table. Estimated FREAD average RMSD and precision on the human and mouse data.** FREAD performance of CDR-H3 structure prediction was validated on the human and mouse data across three CDR-H3 length bins: 5 to 12, 13 and 14, and 15 and 16. For each length bin, ESS cut-offs were selected to achieve an average RMSD better than 3 Å or a coverage greater than 15%. The same ESS cut-offs were selected for both human and mouse data. Precision was defined as the percentage of FREAD predictions within 3 Å over the total number of predictions within the ESS cut-off.
(DOCX)

**S2 Table. SCALOP annotation of Ig-seq data.** Annotation was performed on the human and mouse data. The human data contained 5.7 million sequences with CDR-H3 loop lengths of 16 amino acids or shorter. SCALOP predicted CDR-H1 loop shapes in 97.7% of sequences and CDR-H2 loop shapes in 95.4% in the human data. The total number of mouse sequences was ~207 million, of which 99% of CDR-H1 and ~100% of CDR-H2 loop shapes were annotated.
(DOCX)

**S3 Table. BCR repertoires used in structural diversity analysis.** Ig-seq data columns states species origin of Ig-seq data; modelled columns shows the number of BCR repertoire sequences with predicted CDR-H3 structures; Total column shows the total number of sequences in a given BCR repertoire whose CDR-H3 lengths are between 5 and 16 amino acids.
(DOCX)

**S1 Appendix. Species CDR-H3 template usage.**
(DOCX)

**S2 Appendix. Patterns of CDR-H3 cluster usage.**
(DOCX)

**S3 Appendix. FREAD performance assessment.**
(DOCX)

## Author Contributions

**Conceptualization:** Aleksandr Kovaltsuk, Charlotte M. Deane.

**Data curation:** Aleksandr Kovaltsuk.

**Formal analysis:** Aleksandr Kovaltsuk.

**Funding acquisition:** Charlotte M. Deane.

**Investigation:** Aleksandr Kovaltsuk.

**Methodology:** Aleksandr Kovaltsuk, Charlotte M. Deane.

**Project administration:** Charlotte M. Deane.

**Resources:** Charlotte M. Deane.

**Software:** Aleksandr Kovaltsuk.

**Supervision:** Sebastian Kelm, James Snowden, Charlotte M. Deane.

**Visualization:** Aleksandr Kovaltsuk, Claire Marks.

**Writing – original draft:** Aleksandr Kovaltsuk, Wing Ki Wong.

**Writing – review & editing:** Matthew I. J. Raybould, Claire Marks, Sebastian Kelm, James Snowden, Johannes Trück, Charlotte M. Deane.

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
