## [Decision Letter · Decision Letter 0]

28 Nov 2019

Dear Dr Deane,

Thank you very much for submitting your manuscript, 'Structural Diversity of B-Cell Receptor Repertoires along the B-cell Differentiation Axis in Humans and Mice', to PLOS Computational Biology. As with all papers submitted to the journal, yours was fully evaluated by the PLOS Computational Biology editorial team, and in this case, by independent peer reviewers. The reviewers appreciated the attention to an important topic but identified some aspects of the manuscript that should be improved.

We would therefore like to ask you to modify the manuscript according to the review recommendations before we can consider your manuscript for acceptance. Your revisions should address the specific points made by each reviewer and we encourage you to respond to particular issues Please note while forming your response, if your article is accepted, you may have the opportunity to make the peer review history publicly available. The record will include editor decision letters (with reviews) and your responses to reviewer comments. If eligible, we will contact you to opt in or out.raised.

- Supporting Information uploaded as separate files, titled 'Dataset', 'Figure', 'Table', 'Text', 'Protocol', 'Audio', or 'Video'.

We hope to receive your revised manuscript within the next 30 days. If you anticipate any delay in its return, we ask that you let us know the expected resubmission date by email at ploscompbiol@plos.org.

Sincerely,

Yanay Ofran

Associate Editor

PLOS Computational Biology

Rob De Boer

Deputy Editor

PLOS Computational Biology

[LINK]

Reviewer's Responses to Questions

**Comments to the Authors:**

Reviewer #1: In this study, Kovaltsuk et al. study the structural diversity of BCR repertoires along human and murine B-cell ontogeny.

Structural diversity along B-cell ontogeny has so far not been investigated. To achieve this goal, the authors improved on a previously published software for the structural annotation of antibodies (SAAB  SAAB+) and leveraged two large-scale mouse and human BCR repertoire datasets that comprise different B-cell stages and isotypes. Applying SAAB+ to these datasets, the authors found differences in structural templates used by mice and humans and also a restriction in structural diversity across B-cell ontogeny. Previously, antibody repertoire analyses were limited to sequence-only analysis. Here, Kovaltsuk and colleagues point the way towards structural analyses on entire repertoires across adaptive immune cell thus complementing sequence-only analyses.

This is a very well executed study and i have only minor comments:

- Methods:

-- It has not really become clear to this reviewer what the advancement of saab+ over saab is?

-- Can the authors make this more explicit and dedicate a larger portion on this also to the main text. Can the authors also indicate ho how much time it took to annotate the two datasets (plus hardware requirements).

- Wording:

-- line 36: system(without s)

-- for example line 42: in this work, the authors mix paratope with CDR. While the CDRs contain the paratope, the paratope is mostly one a portion of the CDR and a function of the epitope. Thus, it would be great if the authors could adjust ALL relevant portions of the text accordingly.

-- line 142: the authors state "these results confirm a structural basis for self-tolerance". Can the authors explain this further or remove this sentence from the manuscript?

- Data availability: Has the structural annotation of the two datasets also been deposited? If not, it would be great if the authors could do so so that other groups can build off of their efforts.

- Results

-- The authors performed subsampling experiments to account for the smaller number of PC (antigen-experienced) sequences. However, in antigen-experienced repertoires, the sequence similarity among populations will also likely be higher. Have the authors accounted for that? Or, in other terms, to what extent is the restriction in structural diversity a consequence of the increase sequence similarity?

-Discussion

-- the authors state that structural diversity becomes restricted with B cell ontogeny. Would this mean that the recognition potential become more restricted as well? What does this mean for the memory response which is supposed to be a trade-off between increased diversity and increased affinity?

- Figure aesthetics:

-- the figures are fine but not the most attractive. I don't have anything specific against them but if the authors could dedicate a few hours to making them a bit more visually pleasing, the readers would be very thankful i think.

Reviewer #2: N/A. Withdrawal due to late-perceived conflict of interest per PLoS policy.

Reviewer #3: The authors analyzed heavy chain sequences of human and mouse B-cell repertoires

that were obtained from Observed Antibody Space database. They used antibody

modelling tools (SCALOP and FREAD) to annotate H1, H2 and H3 loops by their

putative structure/conformation. Using the structural annotation the authors

showed that paratopes in naive B-cell populations are structurally distinct from

paratopes in mature B-cells. Furthermore, they observed different paratope

usage between human and mice.

Overall, this is a well executed work that will be appreciated by

researchers involved in immunotherapy and antibody engineering.

I would like to raise following points:

1. Due to the limitation of modelling tools the H3 loop structure predictions

were carried out only for loops shorter than 16 amino acids. As a result less than

50% of observed human heavy chain H3 loops were annotated. The authors should

discuss what impact (if any) may the missing annotations have on the

conclusions regarding changes of paratope usage upon B-cell differentiation.

2. What precluded a similar analysis of light chain sequences? Could authors

discuss what can we expect: similar findings or, on the contrary, learn

something about distinct roles of heavy and light chains?

**Have all data underlying the figures and results presented in the manuscript been provided?**

Reviewer #1: Yes

Reviewer #2: Yes

Reviewer #3: Yes

PLOS authors have the option to publish the peer review history of their article (what does this mean?). If published, this will include your full peer review and any attached files.

Reviewer #1: No

Reviewer #2: No

Reviewer #3: No

---

## [Decision Letter · Decision Letter 1]

7 Jan 2020

Dear Dr Deane,

We are pleased to inform you that your manuscript 'Structural Diversity of B-Cell Receptor Repertoires along the B-cell Differentiation Axis in Humans and Mice' has been provisionally accepted for publication in PLOS Computational Biology.

In the meantime, please log into Editorial Manager at https://www.editorialmanager.com/pcompbiol/, click the "Update My Information" link at the top of the page, and update your user information to ensure an efficient production and billing process.

One of the goals of PLOS is to make science accessible to educators and the public. PLOS staff issue occasional press releases and make early versions of PLOS Computational Biology articles available to science writers and journalists. PLOS staff also collaborate with Communication and Public Information Offices and would be happy to work with the relevant people at your institution or funding agency. If your institution or funding agency is interested in promoting your findings, please ask them to coordinate their releases with PLOS (contact ploscompbiol@plos.org).

Thank you again for supporting Open Access publishing. We look forward to publishing your paper in PLOS Computational Biology.

Sincerely,

Yanay Ofran

Associate Editor

PLOS Computational Biology

Rob De Boer

Deputy Editor

PLOS Computational Biology

Reviewer's Responses to Questions

**Comments to the Authors:**

Reviewer #1: The authors have addressed all reviewer comments and I congratulate the authors on such a well-executed study.

Reviewer #3: The authors addressed all concerns that I raised.

**Have all data underlying the figures and results presented in the manuscript been provided?**

Reviewer #1: Yes

Reviewer #3: Yes

PLOS authors have the option to publish the peer review history of their article (what does this mean?). If published, this will include your full peer review and any attached files.

Reviewer #1: No

Reviewer #3: No

---

## [Editor Report · Acceptance letter]

7 Feb 2020

PCOMPBIOL-D-19-01719R1 

Structural Diversity of B-Cell Receptor Repertoires along the B-cell Differentiation Axis in Humans and Mice

Dear Dr Deane,

I am pleased to inform you that your manuscript has been formally accepted for publication in PLOS Computational Biology. Your manuscript is now with our production department and you will be notified of the publication date in due course.

With kind regards,

Laura Mallard
